# Heterogeneous integration for on-chip quantum photonic circuits with single quantum dot devices

Marcelo Davanco [1], Jin Liu[1,2,3], Luca Sapienza [1,4], Chen-Zhao Zhang[5],
José Vinícius De Miranda Cardoso[1,6], Varun Verma[7], Richard Mirin[7], Sae Woo Nam[7], Liu Liu[5]
& Kartik Srinivasan[1]

Single-quantum emitters are an important resource for photonic quantum technologies, constituting building blocks for single-photon sources, stationary qubits, and deterministic quantum gates. Robust implementation of such functions is achieved through systems that provide both strong light–matter interactions and a low-loss interface between emitters and optical fields. Existing platforms providing such functionality at the single-node level present steep scalability challenges. Here, we develop a heterogeneous photonic integration platform that provides such capabilities in a scalable on-chip implementation, allowing direct integration of GaAs waveguides and cavities containing self-assembled InAs/GaAs quantum dots—a mature class of solid-state quantum emitter—with low-loss $Si_3N_4$ waveguides. We demonstrate a highly efficient optical interface between $Si_3N_4$ waveguides and single-quantum dots in GaAs geometries, with performance approaching that of devices optimized for each material individually. This includes quantum dot radiative rate enhancement in microcavities, and a path for reaching the non-perturbative strong-coupling regime.

[1] Center for Nanoscale Science and Technology, National Institute of Standards and Technology, Gaithersburg, MD 20899, USA. [2] School of Physics, Sun-Yat Sen University, Guangzhou 510275, China. [3] Maryland NanoCenter, University of Maryland, College Park, MD, USA. [4] Department of Physics & Astronomy, University of Southampton, Southampton S017 1BJ, UK. [5] South China Academy of Advanced Optoelectronics, Science Building No. 5, South China Normal University, Higher-Education Mega-Center, Guangzhou 510006, China. [6] Federal University of Campina Grande, Campina Grande, Brazil. [7] National Institute of Standards and Technology, Boulder, CO 80305, USA. Correspondence and requests for materials should be addressed to M.D. (email: marcelo.davanco@nist.gov) or to J.L. (email: liujin23@mail.sysu.edu.cn) or to K.S. (email: kartik.srinivasan@nist.gov)

One of the principal avenues for photonic quantum information processing (QIP) relies on single-photon qubits, with which near-unity fidelity operations can in principle be reached[1]. A tall hurdle towards efficient implementations of single-photon QIP is the difficulty in achieving single-photon nonlinearities for implementing deterministic qubit operations. While measurement-based computation with linear optical networks is a viable alternative[2], the large resource overhead necessary to boost the success rate of non-deterministic gates, and ultimately to significantly scale the size of such systems, is technically very challenging. Indeed, the great level of scalability and stability afforded by photonic integrated circuits has enabled many demonstrations of small-scale quantum computation, simulation, and metrology through this approach[3, 4]; however, scaling such systems towards larger experiments[5] is severely limited by system inefficiencies. In circuits that are, by and large, composed of purely passive elements such as waveguide arrays, phase delays, and beamsplitters, a combination of small photon flux at the circuit input, passive losses in the circuit, and inefficient detection at the output leads to unrealistically long experimental time-scales[6].

In this context, the introduction of solid-state single-quantum emitters as functional elements within such photonic circuits can enable significant scaling of on-chip QIP, in two complementary ways. First, by acting as chip-integrated on-demand, bright sources of indistinguishable single-photons, these elements can significantly boost the photonic flux available for interference experiments, thereby enabling the investigation of significantly more complex quantum computing circuits that rely on post-selection[6]. These sources would also enable single-photon-level investigation of a variety of physical processes available in chip-based nanophotonic and nanoplasmonic structures, such as Kerr nonlinearities[7] and optomechanical interactions[8]. Second, single-emitters strongly coupled to on-chip cavities provide a path towards single-photon nonlinearities[9], and enable deterministic quantum operations through cavity quantum electrodynamics (CQED) within a quantum network formed by a photonic integrated circuit[10].

Towards these goals, we have developed a scalable, integrated, heterogeneous III–V/silicon photonic platform to produce photonic circuits based on $Si_3N_4$ waveguides that directly incorporate GaAs nanophotonic devices, such as waveguides, ring resonators,

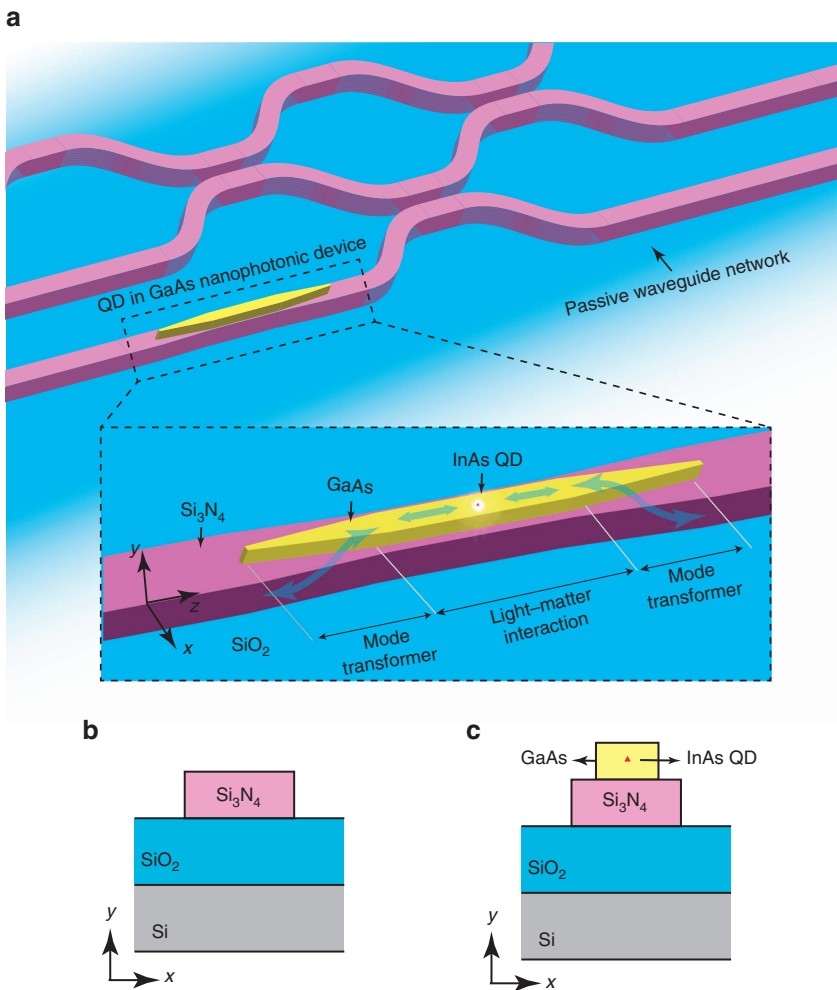

**Fig. 1** Principle of operation and device geometry. **a** Conceptual quantum photonic circuit composed of a passive waveguide network with a directly integrated GaAs nanophotonic device (exemplified by a nanowaveguide) containing a single quantum dot. A zoomed-in image of the GaAs device region (inside the dashed boundary box) shows details of the geometry and operation principle of the hybrid photonic integration platform. The light–matter interaction section of the device promotes efficient coupling between a confined electromagnetic field (in this case, a wave confined in a GaAs nanowaveguide) and a single-InAs QD embedded in the GaAs. Adiabatic mode transformers allow light from the QD in the light–matter interaction region to be efficiently transferred to a $Si_3N_4$ waveguide, and, conversely, also allow the QD to be accessed efficiently with resonant light guided by the $Si_3N_4$ waveguide. **b**, **c** Cross-sections of passive $Si_3N_4$ and active GaAs waveguides that form the core elements of the integration platform

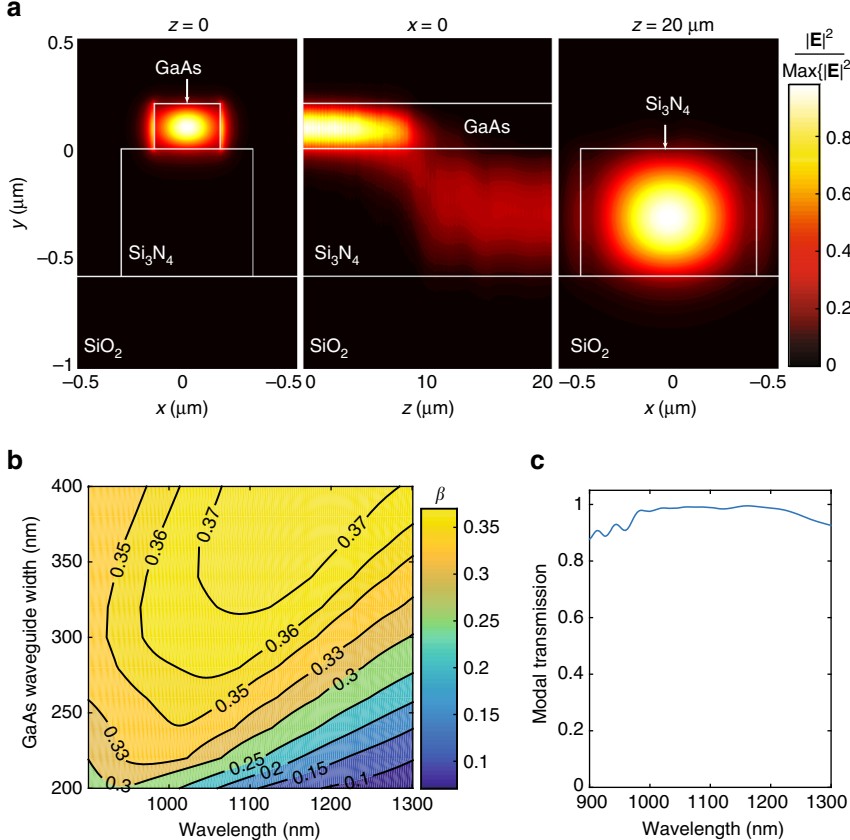

**Fig. 2** Nanophotonic design. **a** Left panel: electric field distribution for the fundamental TE GaAs supermode of the waveguide stack in Fig. 1c, with dimensions specified in the main text. Center panel: electric field distribution across the mode-transformer cross-section, for a GaAs mode launched at $z = 0$. At $z \approx 10\,\mu m$, the GaAs and $Si_3N_4$ guides are phase-matched, and power is efficiently transferred from the top GaAs to the bottom $Si_3N_4$ guide. Right panel: fundamental TE mode of the $Si_3N_4$ waveguide at the end of the mode transformer. **b** Coupling efficiency ($\beta$), as a function of GaAs width and emission wavelength, of photons emitted by a dipole located at $x = 0$ and 74 nm below the top surface, into the GaAs waveguide mode traveling in either the +z or −z direction. **c** Modal power conversion efficiency from the GaAs mode into the $Si_3N_4$ mode in (**a**) as a function of wavelength

and photonic crystals (PhCs), containing single self-assembled InAs/GaAs quantum dots (QDs). Self-assembled InAs/GaAs QDs in GaAs nanophotonic geometries have been used to demonstrate close-to-optimal triggered single-photon emission[11, 12], spin-qubit operation[13], and a variety of strong-coupling CQED systems[14–16]. Importantly, the ability to produce QDs within high index contrast GaAs nanophotonic geometries has been key in many such demonstrations, enabling control of light–matter interactions through high quality factor, small-mode volume optical resonances. Such coupling enables, for example, efficient channeling of the emitted quantum light into a specific optical mode, large Purcell enhancement, and the achievement of the light–matter strong-coupling regime. As a complementary technology, $Si_3N_4$ waveguides offer low-loss propagation with tailorable dispersion and relatively high Kerr nonlinearities, properties which are currently being explored for linear[17] and nonlinear[7] optical signal processing, as well as cavity optomechanics-based measurements[18], down to the quantum level.

As illustrated in Fig. 1a, our integration platform allows the creation of passive, $Si_3N_4$ waveguide-based circuits, which can be used for low-loss routing, distribution, and interference of light across the chip. At select portions of such passive circuits, GaAs waveguide-based nanophotonic geometries containing self-assembled InAs QDs are produced, on top of a $Si_3N_4$ waveguide section. As demonstrated below, such active GaAs geometries can be designed to control light–matter interaction between a single-embedded InAs/GaAs QD and GaAs-confined

propagating waves and localized cavity resonances, while also providing a highly efficient interface between the $Si_3N_4$ circuit and the QD in the GaAs nanophotonic structure, through adiabatic mode transformers. To provide the proof-of-principle that such capabilities are achievable, we produce geometries in which QDs inside GaAs waveguides and microring resonators act as sources of single-photons that are launched with high efficiency into $Si_3N_4$ waveguides. Furthermore, within this platform, we demonstrate effective control of the QD radiative rate by GaAs microring resonators and show the suitability of hybrid micro-disks for the achievement of the light–matter strong-coupling regime in this platform.

Our work extends the application space of a mature, scalable, top-down heterogeneous photonic integrated circuit platform[19] into the quantum realm. While several other hybrid/heterogeneous integration technologies are currently being explored (see Supplementary Note 1 for an extended discussion), our work is unique in allowing independent, flexible, and high-resolution tailoring of both active and passive photonic circuit elements with precise and repeatable, sub-50 nm alignment defined strictly by lithography. Taking advantage of the low losses in the $Si_3N_4$ material, our platform also addresses issues associated with losses that affect the performance of GaAs-based devices.

## Results

**Quantum dot interface design**. While heterogeneous integration of III–V materials for active functionality and silicon-on-insulator

for passive functionality has become widespread in classical integrated photonics[19], design considerations for integrated quantum photonics with single-quantum emitters are significantly distinct. The first distinction is that, because silicon is opaque at wavelengths below 1 μm, it is a poor material for producing low-loss waveguides that carry light from many important solid-state quantum emitters—such as diamond nitrogen vacancy centers, single-laser dye molecules, epitaxial In(Ga)As/GaAs, and GaN QDs, colloidal QDs, defects in SiC, 2D transition-metal dichalcogenides, hexagonal boron nitride, etc[20]. In our platform, we choose to use stoichiometric silicon nitride, which has a wide transparency window, and can accommodate the full range of In(Ga)As/GaAs QDs that have been developed (wavelength ranging from 780 to 1300 nm). The second distinction is that strong light–matter interaction requires a sufficiently strong optical field concentration at the emitter location. Such a requirement is considerably relaxed in hybrid, silicon/III–V integrated classical photonic elements such as lasers and amplifiers, because in such devices a reduced degree of light–matter interaction can be offset by the availability of a high density of emitters that interact with the optical field. Indeed, optical confinement in such structures is typically weak, with guided modes that overlap little with the active III–V gain medium[19]. Importantly, due to the weak vertical optical confinement afforded by such geometries, spontaneous emission modal coupling ($\beta$) factors are typically considerably less than 100%, i.e., III–V emitters contribute considerably <100% of their radiation to the interacting, confined optical field. The interacting optical field is, therefore, a poor conveyor of information about any one single emitter. In contrast, in our platform, the large index contrast between the III–V ($n_{GaAs} \approx 3.5$) and the Si$_3$N$_4$ ($n_{Si_3N_4} \approx 2.0$), can be used to produce optical fields strongly confined in the GaAs material, and which can strongly interact with a single-embedded InAs QD (Supplementary Note 1).

The schematic drawings in Fig. 1b, c respectively show cross-sections of passive and active waveguide sections that form the building blocks of our photonic integration platform. Passive sections consist of Si$_3$N$_4$ ridges with SiO$_2$ and air for bottom and top claddings, respectively, whereas active sections consist of the same Si$_3$N$_4$ ridge, topped by a GaAs ridge containing a single-InAs QD. Active sections are composed of a light–matter interaction geometry (a straight waveguide in the case of Fig. 1a), and adiabatic mode transformer geometries. The light–matter interaction geometry is specifically designed to support guided or localized optical waves that interact strongly with the QD. These can be guided waves of a nano-waveguide, as discussed below, or resonant modes of microring, microdisk, or even 1D PhC resonators. The mode transformer geometries, in turn, are designed to efficiently couple the guided or resonant modes of the light–matter interaction geometry to guided modes of the passive Si$_3$N$_4$ waveguides.

We illustrate these concepts through an example design of a source of single-photons that are launched directly and with high efficiency into a passive Si$_3$N$_4$ waveguide, based on the geometry of Fig. 1a. Here, the light–matter interaction geometry is simply a GaAs waveguide with cross-section in Fig. 1c. The GaAs ridge must support a single transverse-electric (TE) mode, phase-mismatched to the Si$_3$N$_4$ guide. This ensures that the fundamental TE supermode of the waveguide stack is strongly concentrated in the GaAs core, as shown in the left panel in Fig. 2a. The InAs QD must then be made to radiate almost exclusively into the fundamental GaAs supermode, rather than into other guided or unbound modes of the stack. The fraction of the total dipole-emitted power that is coupled to the GaAs mode is the $\beta$-factor, $0 \le \beta \le 1$. $\beta \to 1$ can be achieved for guided modes in waveguides with high refractive index contrasts and sub-

wavelength cross-sections, a result of strong field screening inside the guiding core, that takes place for radiative modes[21]. This has been demonstrated in GaAs nanowires or nanowaveguides surrounded by air[22–24] or encapsulated in SiN[25, 26]. We predict similar performance for a GaAs nanowire on top of a Si$_3$N$_4$ ridge. Assuming a horizontally ($x$) oriented QD electric dipole moment, we use finite difference time domain (FDTD) simulations to compute $\beta$ for the GaAs supermode of an active guide designed for emission wavelengths near 1100 nm. The thicknesses of the GaAs and Si$_3$N$_4$ layers were taken from the wafer stack used for fabrication (see Methods section and Supplementary Note 2). Figure 2b shows a contour map of $\beta$ as a function of wavelength and GaAs waveguide width, for a Si$_3$N$_4$ waveguide thickness of 580 nm and width of 600 nm. For GaAs widths between 300 nm and 400 nm, $0.37 > \beta > 0.35$ for waves traveling in either the $+z$ or $-z$ direction ($0.74 > 2\beta > 0.70$ total) is achievable over ≈100 nm around 1100 nm. Further simulations (not shown) indicate that $\beta$ is robust with respect to the Si$_3$N$_4$ waveguide width, to within several tens of nm. Although $\beta$ is less than the maximum of 0.5 for symmetric emission, we note that both in simulations and in our devices the QD was located at a non-optimal vertical location inside the GaAs. In Supplementary Note 3, we provide similar simulations for an optimized geometry with $\beta > 0.45$ ($2\beta > 0.9$), comparable to those predicted in GaAs nanowires and nanowaveguides[22–24], and in PhC slow-light waveguides[27, 28]. The mode transformer geometry consists of an adiabatic structure in which the widths of the GaAs and Si$_3$N$_4$ waveguides are, respectively, reduced and increased along the $z$ direction. The width tapers are designed such that the two waveguides become phase-matched over some finite length along the mode converter, where power is efficiently transferred from the GaAs to the Si$_3$N$_4$ guide; past the phase-matching length, the taper brings the two guides again away from the phase-matching condition, preventing the power from returning to the top guide. This is illustrated in the middle panel of Fig. 2a, which shows the FDTD-simulated electric field distribution for a transformer in which the GaAs and Si$_3$N$_4$ widths vary linearly from 300 to 100 nm and from 800 to 600 nm, respectively, over a length of 20 μm. Significantly shorter lengths can potentially be achieved with more sophisticated profiles (see, e.g., ref. 29 and references within). Figure 2c shows modal power conversion efficiency from the GaAs mode to the Si$_3$N$_4$ mode (right panel of Fig. 2a) as a function of wavelength (see Methods section for simulation details). Maximum efficiency in excess of 98% is achieved over a >200 nm wavelength range. The geometry is robust to variations of tens of nm in the initial and final widths, well within electron-beam lithography tolerances.

Considering these two combined elements, the maximum efficiency of our ideal single-photon source is $\beta \cdot \eta \approx 0.72$ into both directions of the Si$_3$N$_4$ waveguide, or 36% in either the $+z$ or $-z$ direction. For the optimized design in Supplementary Note 3, efficiency >90% could potentially be achieved. We note that the source here is symmetric, so emission is in either $\pm z$ direction; unidirectional emission can potentially be implemented with an end-mirror or through chiral coupling[30–32]. We furthermore emphasize that the light–matter interaction geometry can take the form of any waveguide-based geometry, such as 1D PhC cavities, or waveguide-coupled microring or microdisk resonators (see below and Supplementary Note 4 for examples), which may provide high $\beta$ through Purcell enhancement. In microring or microdisk resonator geometries, a GaAs bus waveguide must be used to evanescently couple to whispering gallery modes (WGMs); the bus waveguide can in turn be efficiently coupled to an underlying Si$_3$N$_4$ waveguide through mode transformers, as demonstrated in the following sections.

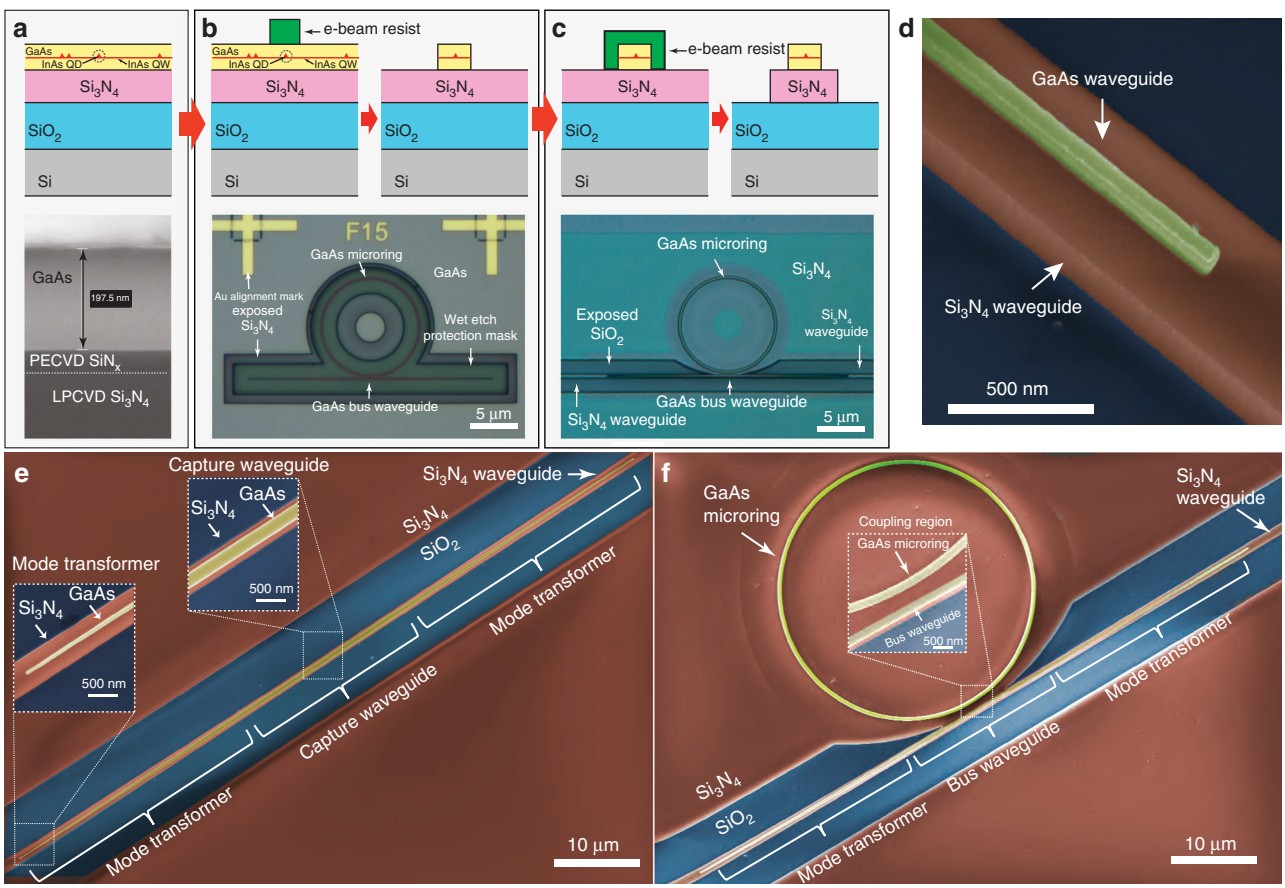

**Fig. 3** Device fabrication. **a** Top: schematic of the bonded wafer stack used in fabrication, consisting of a top III–V layer, containing InAs QDs, that is directly bonded on top of a Si/3 μm SiO₂/550 nm Si₃N₄ stack. Bottom: cross-sectional scanning electron micrograph (SEM) of bonded wafer stack. The ≈30 nm SiNₓ layer was grown on the GaAs wafer surface prior to bonding. **b** Top: GaAs device lithography and etching steps. Bottom: optical micrograph of etched GaAs microring resonator and bus waveguide. Au alignment marks used for registered electron-beam lithography are visible. A wet etch protection resist mask (not depicted in the schematic—Supplementary Note 2) is also visible. **c** Top: Si₃N₄ waveguide lithography (aligned to the previously etched GaAs device) and etching steps. Bottom: optical micrograph of GaAs microring resonator and bus waveguide, and underlying Si₃N₄ waveguide. **d** False-color SEM of tip of mode-transformer geometry, common to both devices in (**e, f**). **e** False-color SEM of fabricated GaAs waveguide (yellow) on top of Si₃N₄ (red) waveguide. Blue regions are exposed SiO₂. Insets show details of the mode transformer end tip and the QD photon capture waveguide. **f** False-color SEM of GaAs microring and bus waveguide, and underlying Si₃N₄ waveguide. Inset shows details of the microring-bus waveguide evanescent coupling region

Moreover, we predict that our platform may allow the creation of high quality factor ($Q \approx 10^6$) microdisk resonators supporting WGMs with volumes of the order of a few cubic wavelengths (Supplementary Note 4). Such devices would be equivalent to those described in ref. [14], with which the strong-coupling CQED regime was achieved. This suggests that our platform may enable the creation of on-chip networks of strongly coupled QD-based coherent CQED systems connected by low-loss waveguides, with which on-chip photonic quantum computation going beyond non-deterministic gate operation might be achieved.

We next describe a fabrication process for devices based on the outlined platform, and experimentally demonstrate two types of on-chip single-photon sources.

**Heterogenous device integration**. We start with the wafer stack shown in Fig. 3a. It consists of a silicon substrate topped by a 3 μm thick thermal oxide layer, a 550 nm layer of stoichiometric Si₃N₄, and an epitaxially grown 200 nm GaAs/AlGaAs stack containing a single layer of InAs quantum dots-in-a-well[33] located 74 nm below the top GaAs surface (details in Supplementary Note 2). As a result of the self-assembled growth, QDs were randomly distributed within this layer, with a density >100/

μm². We point out that the devices reported here contain an ensemble of randomly positioned QDs, rather than a single, spatially isolated QD as illustrated in Fig. 1. Nonetheless, the inhomogeneous broadening of the ensemble ensures that the emission from a single QD can be spectrally isolated by selective excitation, allowing proof-of-principle demonstrations of the many capabilities achievable with individual QDs within our platform.

The hybrid III–V semiconductor/Si₃N₄ stack is produced with a low-temperature, oxygen plasma-activated wafer-bonding procedure[19] detailed in Supplementary Note 2. Following the wafer bonding step, fabrication proceeds as in Fig. 3b, c (optical micrographs of the devices after completion of each step are also shown). An array of Au alignment marks is first produced on top of the GaAs layer via electron-beam lithography followed by metal lift-off. Electron-beam lithography and inductively coupled plasma etching are next used to define GaAs devices aligned to the Au mark array. After cleanup of the etched sample surface, electron-beam lithography referenced to the same Au mark array is performed to define Si₃N₄ waveguide patterns aligned to the previously etched GaAs devices. Reactive ion etching is then used to produce the Si₃N₄ waveguides. As a final step, the chip is cleaved perpendicularly to the Si₃N₄ waveguides >1 mm away

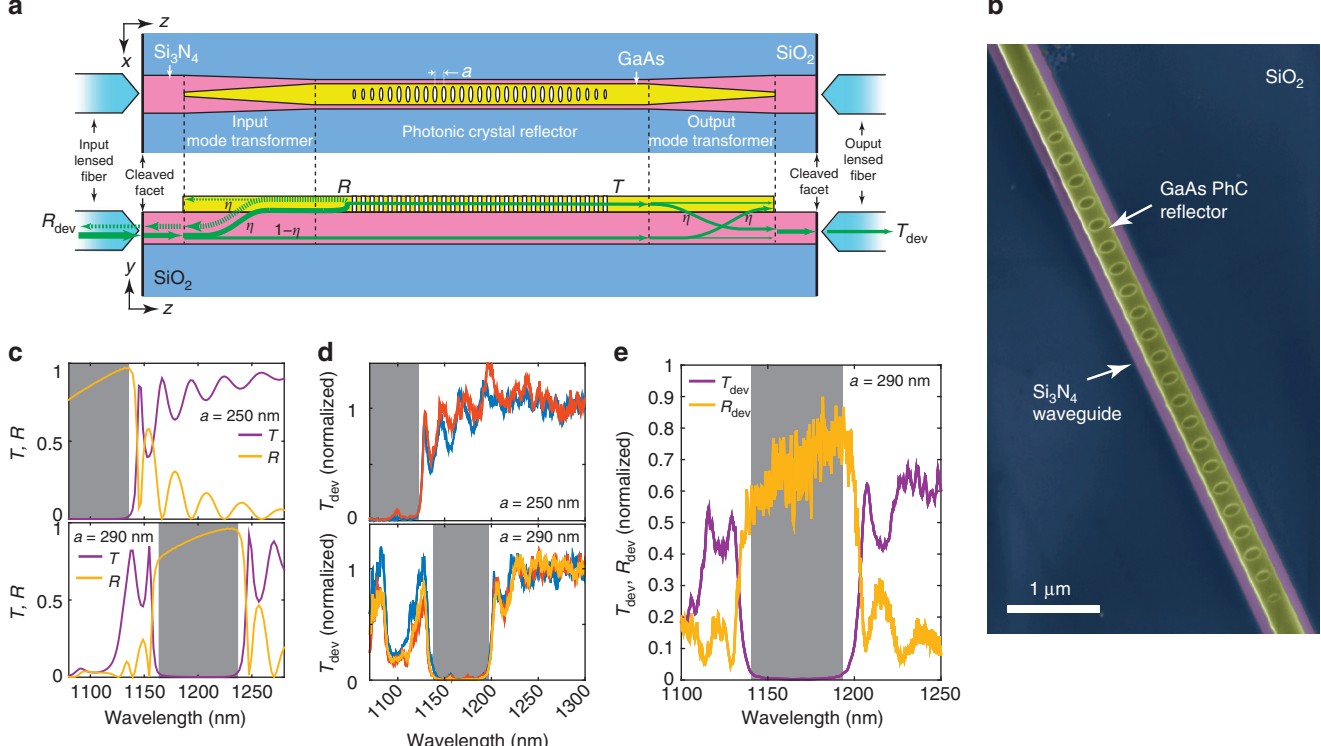

**Fig. 4** Characterizing mode transformer efficiency with a photonic crystal reflector. **a** Schematic of a PhC reflector device. Top: top-view. Bottom: cross-section. Green arrows indicate pathways taken by the optical signal injected at the input port. $R$ and $T$ stand for PhC modal power transmission and reflection spectra, and $T_{dev}$, $R_{dev}$ transmission and reflection spectra through the entire device, including lensed fibers. **b** False-color SEM of fabricated GaAs PhC reflector (yellow) on top of a $Si_3N_4$ (pink) waveguide, on top of exposed $SiO_2$ (blue). **c** FDTD-simulated TE modal transmission ($T$, purple) and reflection ($R$, yellow) spectra as a function of wavelength for the PhC (without mode transformers), for two different lattice constants $a$. **d** Experimental transmission spectra for various PhC reflectors with $a = 250$ nm (top) and $a = 290$ nm (bottom), normalized first to the transmission spectrum of a baseline $Si_3N_4$ waveguide (without GaAs sections), then to the mean transmission at wavelengths between 1250 and 1300 nm. Different colors indicate different devices. **e** Experimental transmission and reflection spectra for a PhC reflector with $a = 290$ nm, normalized to the transmission spectrum of a baseline $Si_3N_4$ waveguide. Gray areas have transmission <−15 dB in (**c**, **d**), <−20 dB in (**e**)

from the GaAs devices, to allow access with optical fibers in the endfire configuration. Before cleaving, 168 devices were produced, with a >80% overall yield considering just device geometry. Features as small as 50 nm were achieved in the GaAs layer, and alignment accuracy on the order of a few tens of nm between the top and bottom waveguides was typically observed. We point out that, although here we had no control over QD location within the fabricated GaAs devices, we have specifically tailored our fabrication sequence to allow seamless incorporation of positioning techniques capable of spatially mapping QDs with respect to the Au marks[34–36].

Figure 3d is a false-color scanning electron micrograph (SEM) of a fabricated stacked-waveguide structure, corresponding to the tip of a mode transformer section. GaAs, $Si_3N_4$, and $SiO_2$ are colored in yellow, red, and blue, respectively. Figure 3e, f show SEMs of two types of fabricated devices, with different emission capture geometries. In Fig. 3e, the capture structure is a straight waveguide as discussed above. The insets show details of the capture and mode transformer sections. In Fig. 3f, the capture structure is a GaAs microring resonator that is evanescently coupled to a bus waveguide with mode transformers, with the same geometry as in Fig. 3e. Here, QD emission coupled to WGMs of the GaAs microring are outcoupled through the bus waveguide (coupling region shown in the inset), and then transferred to the $Si_3N_4$ guide via the mode transformers. We next describe optical measurements done to characterize the photonic performance of the fabricated devices.

**Mode transformer characterization**. Two important parameters common to all types of devices are the mode transformer efficiency $\eta$ and the external coupling efficiency $\eta_{ext}$. The first determines, together with the $\beta$-factor, the efficiency of the interface between the QD-containing GaAs layer and the passive waveguide circuit. The latter is the efficiency with which the device can be accessed from off-chip, ultimately determining the absolute power available for detection.

We estimate the mode transformer $\eta$ via transmission spectroscopy of a third type of device fabricated within our platform, a waveguide-coupled PhC reflector, schematically shown in Fig. 4a. The PhC is a ≈300 nm wide GaAs waveguide into which a periodic 1D array of elliptical holes is etched, with lattice constant $a$. Major and minor hole radii are kept constant over 19 lattice constants at the center, then reduced linearly over 5 constants at the two ends of the array (to minimize radiation losses). The false-color SEM in Fig. 4b illustrates the type of high-resolution GaAs devices achievable within our platform. The periodic hole array defines a photonic bandgap for the TE-polarized GaAs mode on the left panel of Fig. 2a, which is strongly reflected by the PhC at bandgap wavelengths. Figure 4a describes the PhC reflector operation. Light is launched into the $Si_3N_4$ waveguide using a lensed optical fiber aligned to its cleaved facet, then transferred with efficiency $\eta$ to the GaAs waveguide via the input mode transformer. At bandgap wavelengths, the GaAs-guided light is reflected with reflectivity $R$ by the PhC, then transferred back into the $Si_3N_4$ waveguide via the input transformer, with efficiency $\eta$.

Simulated TE GaAs mode power transmission ($T$) and reflection ($R$) spectra are shown in Fig. 4c, for PhCs with $a = 250$ nm and $a = 290$ nm and dimensions estimated by SEM from fabricated devices. Photonic bandgaps are evidenced by high reflectivity, high transmission extinction spectral regions marked in gray. We emphasize that $R$ and $T$ are spectra for the GaAs-confined modes, i.e., they do not include effects due to the mode transformers. We nevertheless observe, in Fig. 4d, similar features experimentally, which suggests spectrally broad mode transformer operation consistent with Fig. 2c. The experimental setup used is described in the Methods section and Supplementary Fig. 8. Room-temperature characterization is adequate to assess the low-temperature performance, given the spectrally broadband nature of the elements involved and the expected thermo-optic shift of GaAs. Figure 4d shows normalized experimental TE-polarized transmission spectra for various fabricated devices with either $a = 250$ nm or $a = 290$ nm. Consistent spectral features achieved across many devices indicate that our photonic integration platform is scalable. Figure 4e shows a typical PhC reflectivity ($R_{dev}$) peak, obtained for one of the $a = 290$ nm devices, spectrally aligned with the transmission extinction region. The >20 dB ($\approx25$ dB at bandgap center) extinction highlighted in gray indicates highly efficient coupling from the $Si_3N_4$ access waveguide into the GaAs layer, since light not transferred to the GaAs is not reflected by the PhC. As described in the Methods section, the photonic bandgap extinction can be used to obtain a lower bound for the mode transformer efficiency $\eta$. For a typically observed 20 dB extinction, $\eta > 90\%$, conservatively. For the peak extinction of $\approx25$ dB, $\eta > 94\%$.

To determine the external coupling efficiency $\eta_{ext}$, we took the transmitted power spectrum of a blank $Si_3N_4$ waveguide (i.e., with no GaAs devices) and normalized it by the supercontinuum source power spectrum. Assuming identical waveguide facets on both chip edges, $\eta_{ext} = 0.23 \pm 0.03$ over the 1100 to 1300 nm wavelength range, across three different devices (uncertainties are propagated single standard deviations. See Supplementary Fig. 8b for transmission spectra). To verify this, we estimated a mode-mismatch coupling efficiency $\eta_{facet} \approx 26\%$ between the $Si_3N_4$ waveguide mode and a Gaussian beam with 2.5 μm diameter, consistent with the nominal lensed fiber spot-size diameter. The small difference between the experimental coupling efficiency and the calculated value suggests that propagation losses in the waveguide are relatively small. Indeed, in Supplementary Note 5, propagation losses of $\approx1.1$ dB/cm are estimated from a $Si_3N_4$ microring resonator transmission spectrum.

**Quantum dot coupling to waveguides.** We next investigated QD emission coupling in our devices via photoluminescence (PL) measurements at cryogenic temperatures. In our setup, shown in Supplementary Fig. 5, devices were placed inside a liquid Helium flow cryostat, kept fixed on a copper mount connected to the cold finger. Testing temperatures ranged between 7 and 30 K. A microscope system allowed individual devices to be visually located and optically pumped with laser light focused through a microscope objective. PL was collected by aligning a lensed fiber (mounted on a xyz nanopositioning stage inside the cryostat) to the corresponding $Si_3N_4$ waveguide facet. The collected PL was either sent to a grating spectrometer equipped with a liquid nitrogen cooled InGaAs detector array for spectrum measurements, or towards a pair of amorphous WSi superconducting nanowire single-photon detectors (SNSPDs)[37] for time-correlated

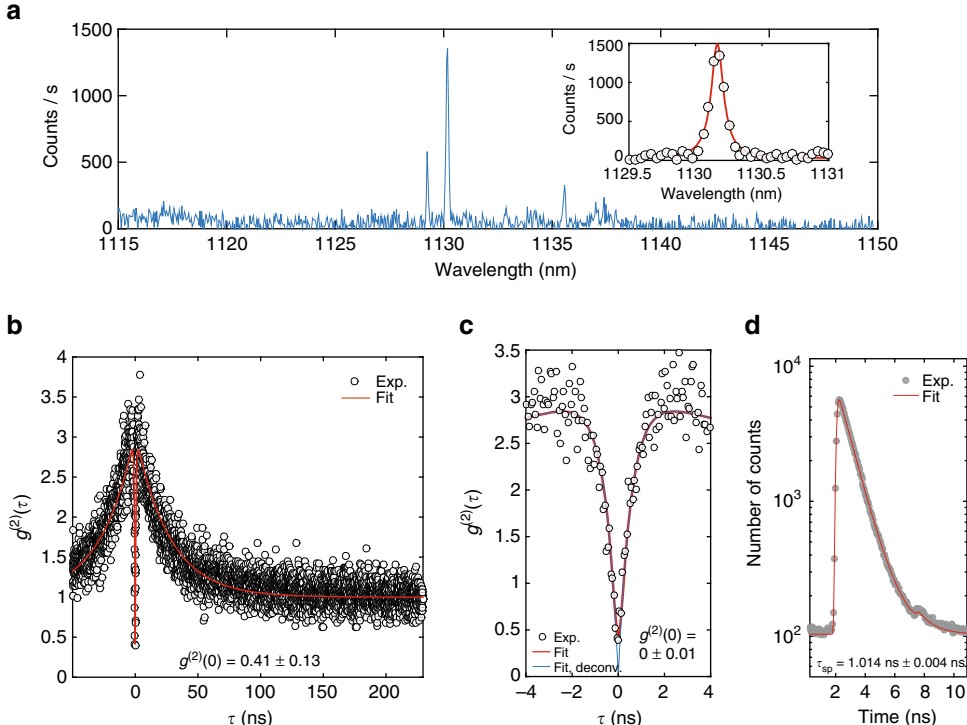

**Fig. 5** Quantum dot-waveguide coupling. **a** Photoluminescence spectrum for a single QD inside a GaAs waveguide as in Fig. 3e, pumped with 1061 nm wavelength laser light. The PL is transferred to the bottom $Si_3N_4$ waveguide, and collected with a lensed optical fiber inside of a Liquid Helium flow cryostat (Supplementary Note 6). Sharp lines are exciton transitions from a single QD. Inset: Fit of PL peak at 1130.18 nm. **b** Second-order correlation as a function of time delay $\tau$ for the 1130.18 nm line. Circles mark experimental data, red line is a fit (Methods section and Supplementary Note 7). **c** Zoom-in of **b** near $\tau = 0$. The blue curve and quoted $g^{(2)}(0)$ are obtained from the red fit by deconvolving the detection time-response. Uncertainties for $g^{(2)}(0)$ are 95% fit confidence intervals (two standard deviations). **d** Photoluminescence decay trace for the 1130.18 nm line. Gray dots are experimental data, the red line is a fit with a monoexponential function with lifetime $\tau_{sp}$. The uncertainty is obtained from the fit and corresponds to a single standard deviation

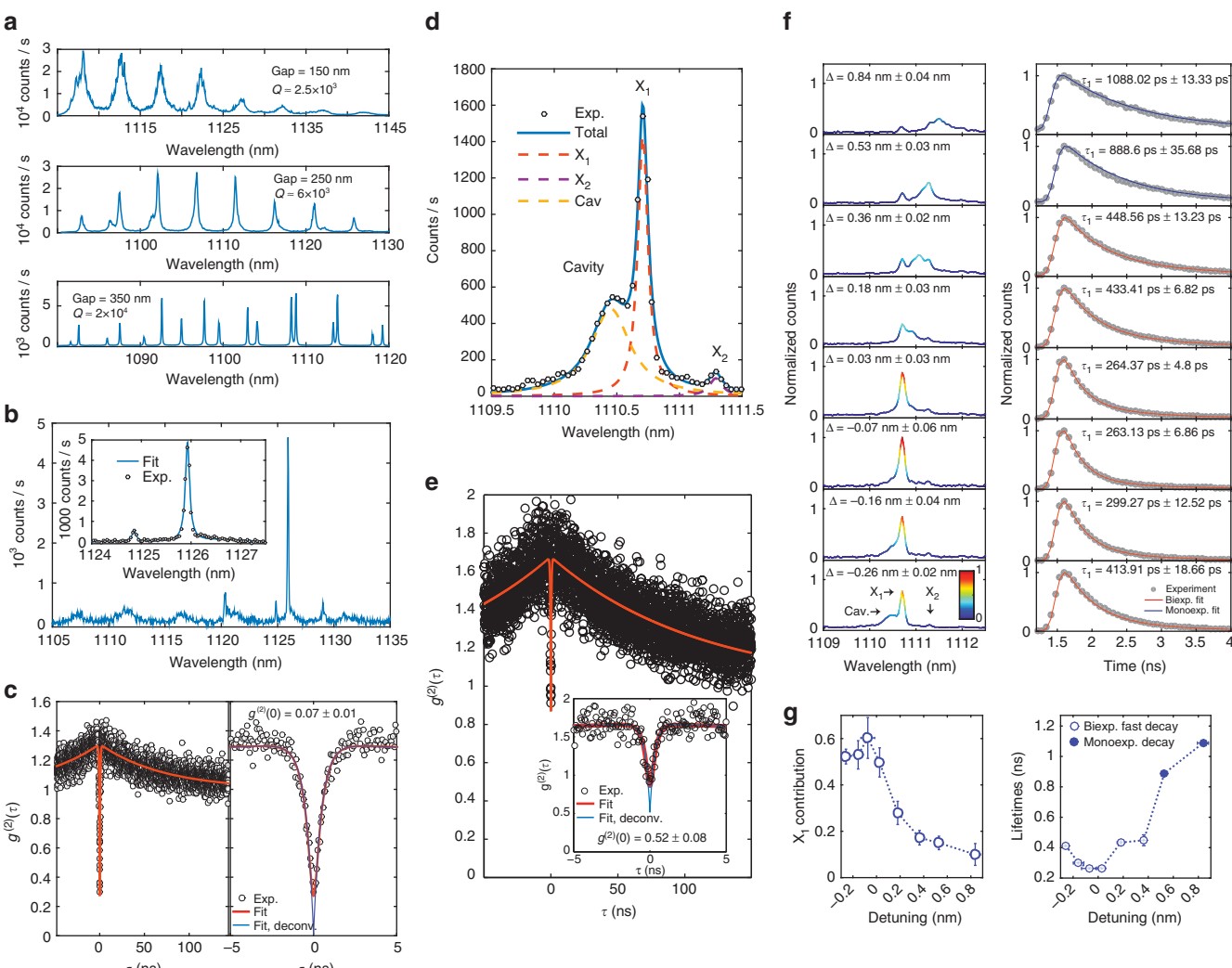

**Fig. 6** Quantum dot-cavity coupling. **a** Photoluminescence (PL) spectra as a function of wavelength from a QD ensemble pumped with laser light at 975 nm, emitting inside three different GaAs microring resonators. Peaks are whispering gallery modes (WGMs) with quality factor Q, which increases with the microring-bus waveguide gap width. **b** PL spectrum for a single QD coupled to a $Q \approx 1.1 \times 10^4$ WGM. Inset: fit of cavity-coupled QD emission near 1126 nm. **c**, Left: second-order correlation $g^{(2)}(\tau)$ for the 1126 nm exciton line in **b**. Right: close-up near $\tau = 0$. Circles are experimental data, red lines are a fit. The blue curve and quoted $g^{(2)}(0)$ are obtained from the red fit by deconvolving the detection time-response. **d** PL spectrum for a single QD in a microring, coupled to a $Q \approx 6 \times 10^3$ WGM. Circles: experimental data. Blue continuous line: fit. Dashed lines: fitting Lorentzians for the cavity and two excitons, $X_1$ and $X_2$. **e** $g^{(2)}(\tau)$ for $X_1$ in **d**. Inset: close-up near $\tau = 0$. **f**, Left panel: PL spectra for varying spectral detuning $\Delta$ between $X_1$ and the cavity. $\Delta$ is obtained from fits as in **d**. All spectra are normalized to the intensity maximum at $\Delta \approx -0.07$ nm. The color scale indicates normalized intensity. Right panel: corresponding $X_1$ photoluminescence decay curves. Gray dots are experimental data, red (blue) lines are biexponential (monoexponential) decay fits. For biexponential fits, $\tau_1$ is the fast lifetime. **g**, Left panel: integrated intensity as a function of $\Delta$ for the filtered $X_1$ exciton contribution to the the PL spectra in **f**, obtained from Lorentzian fits as in **d**, normalized by the integrated intensity of the full fitted spectrum. Right panel: decay lifetimes for the fits in **f**, as a function of $\Delta$. Open blue circles are the fast biexponential decay lifetimes, closed blue circles are the monoexponential decay lifetimes. Uncertainties for $g^{(2)}(0)$, $\Delta$ and the $X_1$ magnitude are 95% fit confidence intervals (two standard deviations). Lifetime uncertainties are single standard deviations from the exponential decay fits

single-photon counting (TCSPC) measurements. We note that the high density QD population in our sample displayed a wide inhomogeneously broadened spectrum, with ensemble s-shell and p-shell peaks located approximately at 1100 and 1060 nm, respectively.

We first investigated QD emission inside the basic hybrid device, a ≈300 nm wide, 10 μm long GaAs waveguide with 20 μm long mode transformers, coupled to a 800 nm wide $Si_3N_4$ waveguide. Figure 5a shows the PL spectrum collected at a temperature of ≈7 K for a device pumped at $\lambda = 1061$ nm (p-shell) with a tunable external-cavity diode laser (ECDL). Sharp spectral lines are excitonic complexes of individual QDs.

A ≈700 pm full-width at half-maximum (FWHM) bandpass grating filter was used to spectrally isolate the line at 1130.18 nm in Fig. 5a, and a Hanbury Brown and Twiss (HBT) setup was used to measure the autocorrelation $g^{(2)}(\tau)$, in Fig. 5b, c. The values $g^{(2)}(0) = 0.41 \pm 0.13$ obtained for the raw data, and $g^{(2)}(0) = 0 \pm 0.13$ obtained by taking into account the ≈129 ps time resolution of our TCSPC system (Methods section), indicate that the QD in the GaAs device acts as source of single-photons that are directly launched into a $Si_3N_4$ waveguide. $g^{(2)}(0)$ uncertainties quoted here and below are 95% fit confidence intervals (two standard deviations). Bunching at $\tau \approx \pm 2$ ns suggests QD blinking, as observed with quasi-resonant (p-shell)

excitation in ref. [38], and could be related to coupling of the radiative excited state to dark states. Our fits were done with a function that models coupling of a two-level system to a single dark state[39].

Lifetime measurements for the same QD line were next performed by modulating the ECDL pump light with an electro-optic modulator (EOM; see Methods section and Supplementary Note 6). The decay curves shown in Fig. 5d were fitted with a single exponential function, revealing a lifetime $\tau_{sp} = 1.014$ ns $\pm 0.004$ ns (lifetime uncertainties here and below are from the fit and correspond to one standard deviation). Assuming a fiber-to-chip coupling efficiency of 22%, and a coupler efficiency $\eta = 98\%$, we estimate a QD-waveguide coupling parameter $\beta = 0.20 \pm 0.07$ (uncertainty from propagated errors in the optical characterization of the measurement system, corresponding to one standard deviation. See Methods for details). This value, though appreciable, is less than the theoretical maximum of 0.37. This discrepancy could be attributed to non-optimal QD position and electric dipole moment orientation.

**Weak-coupling cavity QED.** We next investigated cavity effects on the radiative rate of single QDs coupled to WGMs of GaAs microring resonators (Fig. 3f). The devices consisted of 20-μm diameter microrings formed by ≈300 nm wide waveguides, evanescently coupled to ≈300 nm wide GaAs bus waveguides spaced by gaps of varying dimensions. In this scheme, light from QDs inside the ring is outcoupled through the bus waveguide and then transferred to the $Si_3N_4$ waveguide via the mode transformers. Figure 6a shows PL spectra for three different resonators, with coupling gaps of 150, 250, and 350 nm, pumped at high intensities with 975 nm laser light (resonant with the quantum well transitions). Peaks are PL from the QD ensemble coupled to WGMs. Quality factors for devices with the gap spacings of 150, 250, and 350 nm are $2.5 \times 10^3$, $6 \times 10^3$, and $2 \times 10^4$. The increased $Q$ for larger gaps is due to a decreased cavity-bus waveguide coupling, indicating that the geometrical control afforded by our fabrication platform enables fine control of cavity outcoupling rates. Pumping one of the $Q \approx 1.1 \times 10^4$ microresonators at 1058 nm (p-shell) allowed observation of the single QD excitonic line at 1125.92 nm in Fig. 6b, which was coupled to one of the cavity's WGMs. Background emission, likely from other QDs and (multi)excitonic complexes in the active material, is also observed in the different WGMs. Figure 6c indicates the cavity-coupled QD acts as a single-photon source with $g^{(2)}(0) = 0.28 \pm 0.01$ ($g^{(2)}(0) = 0.07 \pm 0.01$ adjusted for detection time resolution).

We next demonstrated tunable control of Purcell radiative rate enhancement in a device with $Q \approx 6 \times 10^3$, at a fixed temperature of ≈7 K. Pumping at $\lambda = 1065$ nm (p-shell) allowed us to observe the cavity-mode-coupled single QD exciton line $X_1$ in Fig. 6d, as well as a cavity-detuned exciton $X_2$. For the $X_1$ line, as seen in Fig. 6e, $g^{(2)}(0) = 0.72 \pm 0.08 > 0.5$ ($g^{(2)}(0) = 0.52 \pm 0.08$ adjusted for detection time resolution), due to background emission from the cavity mode, which was transmitted by the bandpass filter introduced before detection. Indeed, based on the fit shown in Fig. 6d, cavity emission corresponds to ≈45% of the filtered light intensity. To tune the cavity with respect to the QD exciton, we used the nitrogen gas-tuning mechanism of ref. [40]. A small amount of gaseous $N_2$ is introduced in steps into the cryostat, and gettering at the GaAs surfaces red-shifts the cavity resonance by a small amount at each step. This is observed in the left panel in Fig. 6f, where the PL spectrum of the cavity-coupled QD exciton ($X_1$) is seen to grow in intensity as its spectral (wavelength) detuning $\Delta$ from the cavity center tends to zero. The variation in intensity comes together with a variation in the exciton lifetime, evident in the corresponding decay curves on the right panel of

Fig. 6f. Biexponential fits to the decay data (monoexponential for $\Delta \approx 0.53$ nm and $\Delta \approx 0.84$ nm) are also shown. The detuning-dependent variations in $X_1$ intensity and decay lifetime are summarized, respectively, in the left and right panels in Fig. 6g, evidencing high-resolution, strong control of the exciton radiative rate via cavity coupling achieved in our platform. Further details on PL spectrum and decay fitting and assignment of lifetimes are given in Supplementary Note 7. Comparing with the ≈1 ns lifetime in the waveguide, we can extract a maximum radiative rate enhancement factor of ≈4 for the QD. From the calculated WGM mode volume $V_{eff} = 75.5(\lambda/n_{GaAs})^3$ ($n_{GaAs}$ is the GaAs refractive index) and the experimental $Q = 6 \times 10^3$, we expect a maximum Purcell Factor $F_p \approx 6$ (Methods section). Though reasonably close to the theoretical value, the lower experimental Purcell factor could be due to non-optimal spatial location and polarization alignment of the QD with respect to the microring mode.

## Discussion

The results presented demonstrate that our platform enables the creation of integrated photonic circuits that incorporate quantum-dot-based devices with complex geometries. As discussed above, further improvements to the single-photon capture efficiency (quantified by the $\beta$-factor) can be achieved through optimized wafer stacks (both $Si_3N_4$ and the GaAs epi-stack) and device geometries. In particular, our platform allows the creation of geometries providing high Purcell radiative rate enhancement where high $\beta$ may be achieved, such as microdisk, microring, or PhC-based cavities and slow-light waveguides. The high reflectivity achieved with our PhC reflectors furthermore suggests a path forward towards unidirectional QD emission in a waveguide. Alternatively, chiral coupling to waveguide modes[30] could also be explored. Strongly coupled QD-cavity systems[14–16] evanescently coupled to a bus waveguide could also be envisioned in our platform.

As mentioned above, our III–V wafers contained a high density of QDs (>100/μm²), randomly distributed across the wafer surface, which led to the deterioration of the purity of our on-chip single-photon sources. It is also possible that the pronounced blinking observed in the autocorrelation traces might stem from interactions between many neighboring QDs. Low-density QD growth constitutes a clear way forward here. In this case, QD positioning techniques such as the one developed in refs. [34, 35] —a technique fully compatible with our fabrication process—become essential. Precise QD location within a nanophotonic structure would also allow $\beta$ and Purcell factor optimization.

The underlying $Si_3N_4$ waveguides demonstrated here provide not only a way to route single-photons with low loss across the chip, but also a means to explore nonlinear optical processes with single photons. For instance, four-wave-mixing-based wavelength conversion of single-photon-level laser light was recently demonstrated in a $Si_3N_4$ microring resonator with cross-sectional dimensions similar to those of our waveguides, and fabricated with the same etch process[7]. This means that the required dispersion profiles and nonlinear coefficients are attainable within our platform. At the same time, passive structures with cross-sections optimized for low propagation losses may also be implemented, for instance with thinner $Si_3N_4$ (Supplementary Note 3) and potentially even with a top oxide cladding, which would also enable lower-loss off-chip coupling to optical fibers. The introduction of elements such as on-chip delay lines, high quality $Si_3N_4$-based filters, and microring add-drops, can also be envisioned.

Our platform is also amenable to further integration with waveguide-based SNSPDs[41]. Finally, the fabrication process can be adapted for materials such as AlN and $LiNbO_3$, which may

enable active electro-optic phase control. We anticipate all of these features will enable a new class of monolithic on-chip devices comprising emission, routing, modulation, and detection of quantum light.

## Methods

**Numerical simulation.** Calculations of waveguide $\beta$-factors is done with FDTD simulations. We simulate a $x$-oriented electric dipole source radiating inside the GaAs ridge of the stacked GaAs/Si$_3$N$_4$ waveguide structure shown in Fig. 1c. The simulation is 3D, and the coupled waveguide structure length is 1 μm. Perfectly matched layers are used to emulate either open regions (air and SiO$_2$ semi-infinite spaces above and below the geometry), or infinite waveguides (in the planes perpendicular to $x$ and $y$). We obtain the steady-state electromagnetic fields at the six boundaries of the simulation window, and compute the total emitted power $P$ by integrating the steady-state Poynting vector through them. At the $+z$ and $-z$ planes, we calculate overlap integrals of the radiated field with the field of the fundamental TE GaAs mode (Fig. 2a left panel, at $\lambda = 1100$ nm). This allows us to determine $\beta$, the fraction of the total emitted power that is carried through the $\pm z$ planes by the GaAs mode.

The mode transformer simulations are also performed with FDTD. We launch the fundamental TE GaAs mode of the waveguide structure in Fig. 1c, shown in the left panel of Fig. 2a, into the mode transformer, at the $z = 0$ plane. We obtain the steady-state electromagnetic fields at the output ($z = 20$ μm) plane on the mode transformer, and calculate the overlap integral between this and the output Si$_3$N$_4$ mode (right panel on Fig. 2a). Dividing it by the launched input power we obtain the mode transformer coupling efficiency $\eta$.

We proceed similarly for the simulation of modal reflectivity and transmissivity for the PhC reflector of Fig. 4a. For reflectivity, we place a field monitor at the $z = 0$ plane, and the source at $z = 100$ nm.

To determine the mode volume $V_{eff}$ used in the Purcell factor estimate, we use $V_{eff} = \int_V d\mathbf{V}\epsilon(\mathbf{r})|\mathbf{E}(\mathbf{r})|^2/\max\{\epsilon(\mathbf{r})|\mathbf{E}(\mathbf{r})|^2\}$, where the volume integral is evaluated over the entire microring resonator. Because the ring radius is large ($R = 10$ μm), we assume the WGM fields across the microring cross-section have the same distribution as the fundamental TE GaAs mode of the left panel of Fig. 2a, and an azimuthal dependence $\exp(i \cdot m\phi)$. Then, $V_{eff} = 2\pi \cdot R \cdot \int_A dA\epsilon(\mathbf{r})|\mathbf{E}(\mathbf{r})|^2/\max\{\epsilon(\mathbf{r})|\mathbf{E}(\mathbf{r})|^2\}$, where $A$ is the cross-sectional waveguide area. The maximum Purcell factor (assuming spatial and polarization alignment of the dipole) is calculated with the expression $F_p = (3/4\pi^2) \cdot Q/V'_{eff}$, where $V'_{eff}$ is the mode volume in cubic wavelengths in the GaAs.

**Experimental determination of mode transformer coupling efficiency.** Power transmission and reflection spectra $T_{dev}$ and $R_{dev}$ are determined experimentally using the setup in Supplementary Fig. 8a. Light from a fiber-coupled supercontinuum laser source is passed through a 3 dB fiber directional coupler and polarization controller, then launched into the input waveguide with a lensed fiber. Transmitted light is collected with another lensed fiber aligned to the output waveguide facet at the opposite edge of the chip, and sent to an optical spectrum analyzer (OSA). Reflected light is captured by the input fiber, and routed to the OSA via the 3 dB splitter.

To estimate a lower bound for $\eta$, we use a simple model to obtain an expression for the transmitted power at the output, $T_{dev}$, as suggested in Fig. 4a. Light launched at the input Si$_3$N$_4$ waveguide is transferred with efficiency $\eta$ into the GaAs guide, whereas a residual $(1 - \eta)$ portion of the original power remains in the Si$_3$N$_4$ guide. Light transferred to the GaAs guide will be reflected with a reflectivity $R$ by the PhC, and transmitted through it with transmissivity $T$. The output mode transformer converts light transmitted through the PhC reflector back into the Si$_3$N$_4$ guide, with efficiency $\eta$. We assume that the residual light that remains in the Si$_3$N$_4$ after the input mode transformer is unaffected by the PhC, after which it is partially transferred with efficiency $\eta$ to the GaAs guide by the output mode converter, and is then lost as radiation at the terminated GaAs structure tip. Light collected by the output lensed fiber thus has two components, one that remains in the Si$_3$N$_4$ guide, and one that is transferred to and from the GaAs guide, and interacts with the PhC reflector. The maximum power collected by the output lensed fiber is $T_{dev}$, with

$$T_{dev} \leq \eta_{ext}\left[\eta^2 T + (1 - \eta)^2 + 2 \cdot \eta(1 - \eta)\sqrt{T}\right]. \quad (1)$$

Inside the square brackets, the first and second terms correspond respectively to light transmitted through the PhC and residual light that remains in the Si$_3$N$_4$ guide, and the third term comes from the interference between the two. The transmitted power for wavelengths in and out of the bandgap region are $T_{dev,in}$ and $T_{dev,out}$, respectively, and we define the extinction ratio $\alpha = \frac{T_{dev,in}}{T_{dev,out}}$. Because experimentally $T_{dev,in}$ is at least one order of magnitude lower than $T_{dev,out}$, we can assume that the PhC transmission at bandgap wavelengths is negligible, so that $T \approx 0$ and

$$\alpha > \frac{(1 - \eta)^2}{\eta^2 T + (1 - \eta)^2 \pm 2 \cdot \eta(1 - \eta)\sqrt{T}} > \frac{(1 - \eta)^2}{\eta^2 + (1 - \eta)^2 + 2 \cdot \eta(1 - \eta)} \quad (2)$$

Isolating $\eta$, we obtain the inequality $\eta^2 + (2 - \alpha)/(\alpha - 1) + 1 < 0$. The minimum root of the quadratic equation is our lower bound for $\eta$. For $\alpha = -20$ dB, as typically observed in our PhC spectra, $\eta > 90\%$, conservatively. For the peak extinction of $\approx 25$ dB, $\eta > 94\%$.

**Experimental determination of external coupling efficiency.** The external coupling efficiency $\eta_{ext}$ includes the chip-to-fiber coupling efficiency and propagation losses in the Si$_3$N$_4$ waveguide leading to the device. We employ the setup of Supplementary Fig. 8a to obtain the transmitted power spectrum of a blank Si$_3$N$_4$ waveguide (i.e., with no GaAs devices). Prior to this measurement, the polarization of the incident light is set to TE by probing a PhC reflector and minimizing the transmitted power over the photonic bandgap with the polarization controller. The lensed fibers are then aligned to the blank Si$_3$N$_4$ waveguide, and the transmission spectrum is recorded. The spectrum is then normalized by the supercontinuum source power spectrum, obtained by bypassing the lensed fibers and the device. The resulting transfer function accounts for insertion losses through the two lensed fibers ($\approx 31\%$), and through the device, $IL_{dev} = \eta_{dev}^{-1} = (\eta_{ext,in} \cdot \eta_{ext,out})^{-1}$. Assuming that the waveguide facets are identical on both edges of the chip, $\eta_{ext,in} = \eta_{ext,out} = \eta_{ext}$, the external coupling efficiency is $\eta_{ext} = \sqrt{\eta_{dev}}$. Supplementary Fig. 8b shows the average measured $\eta_{ext}$ for three different waveguides as a function of wavelength (the red curve and gray area correspond to the mean and standard deviation over the three measurements, respectively). Averaging this curve across the 1100 to 1300 nm wavelength range produces $\eta_{ext} = 0.23 \pm 0.03$ (the uncertainty is obtained by propagating the standard deviations from the three devices). The theoretical mode-mismatch coupling efficiency is calculated with the overlap integral

$$\eta_{facet} = \frac{Re\left\{\iint_S (\mathbf{e}_f \times \mathbf{h}^*) \cdot \hat{z}\, dS \iint_S \left(\mathbf{e} \times \mathbf{h}_f^*\right) \cdot \hat{z}\, dS\right\}}{Re\left\{\iint_S \left(\mathbf{e}_f \times \mathbf{h}_f^*\right) \cdot \hat{z}\, dS\right\} Re\left\{\iint_S (\mathbf{e} \times \mathbf{h}^*) \cdot \hat{z}\, dS\right\}} \quad (3)$$

taken over the cross-sectional area $S$ of the input/output Si$_3$N$_4$ waveguide. Here, $\mathbf{e}$ and $\mathbf{h}$ are the electric and magnetic field components of the fundamental TE Si$_3$N$_4$ input/output waveguide mode (right panel on Fig. 2a), and $\mathbf{e}_f$ and $\mathbf{h}_f$ are the field components of a focused Gaussian beam with a spot size of 2.5 μm. The Gaussian beam spot size is consistent with specifications from the lensed fiber manufacturer. With Eq. (3), we obtain $\eta_{facet} \approx 26\%$ for a 580 nm thick $\times$800 nm wide Si$_3$N$_4$ waveguide, at a wavelength of 1110 nm.

**Second-order correlation measurements and fits.** A HBT setup was used to obtain the second-order correlation function $g^{(2)}(\tau)$ of QD emission upon continuous-wave pumping. In our experiments, histograms of delays between detection events in the two single-photon detectors were measured. We related these histograms to $g^{(2)}(\tau)$ as explained below. We first calculated delay probability distributions $C(\tau)$ by normalizing the delay histograms. Sufficiently far away from zero time delay, $C(\tau) \approx A \exp(-A\tau)$. We took the 1000 longest-delay bins of our histograms and perform a log–log linear fit to obtain $A$. The histograms were then normalized by $A$. For $\tau \approx 0$, $g^{(2)}(\tau) \approx C(\tau)$ (see ref. [42]). The $g^{(2)}(\tau)$ data was modeled with the double-exponential function

$$g^{(2)}(\tau) = 1 + A_1 \exp(\lambda_1 \cdot \tau) + A_2 \exp(\lambda_2 \cdot \tau), \quad (4)$$

with $A_1 + A_2 = -1$. This functional form is expected from a two-level system coupled to a single dark state[39], and describes both antibunching at $\tau = 0$, bunching at some later time delay, and a return to the Poissonian level at $\tau \to \infty$. To take into account the $\sigma \approx 129$ ps time-response of detection system (see below for details), we convolved the $g^{(2)}(\tau)$ above with a normal distribution function $N(\tau, \sigma)$:

$$g_C^{(2)}(\tau) = g^{(2)}(\tau) * N(\tau, \sigma) = 1 + A_1 E_1(\tau) + A_2 E_2(\tau), \quad (5)$$

where

$$E_n(\tau) = \frac{\lambda_n}{2} \exp\left(\frac{\lambda_n \sigma}{2}\right) \left\{ erf\left(-\frac{\tau}{\sqrt{2}\sigma} + \frac{\lambda_n \sigma}{\sqrt{2}}\right) e^{-\lambda_n \tau} + erf\left(\frac{\tau}{\sqrt{2}\sigma} + \frac{\lambda_n \sigma}{\sqrt{2}}\right) e^{\lambda_n \tau} \right\} \quad (6)$$

and $n = 1, 2$. Finally, to account for a Poissonian background, we used[42]

$$g_{C,B}^{(2)}(\tau) = 1 + \frac{1}{(1 + b)^2}\left[g_C^{(2)}(\tau)\right]. \quad (7)$$

The fits shown in the main text were done using $g_{C,B}^{(2)}(\tau)$ above, through a nonlinear least-squares procedure. For the QD in a waveguide of Fig. 5b, c, the background $b$ was used as a fit parameter, while for the cavity-coupled QDs of Fig. 6c, e, $b$ was fixed at values estimated from fits to emission spectra (see below for spectrum fitting procedures). To plot $g^{(2)}(\tau)$ without the effect of the finite timing resolution, we used $\sigma = 0$ in Eq. (6) and used the same fitting parameters.

Uncertainties quoted for $g^{(2)}(0)$ are 95% fit confidence intervals, corresponding to two standard deviations.

**Photoluminescence spectrum fits.** The PL spectra in Fig. 6b, f were fitted with a sum of three Lorentzians, representing the cavity and two excitons, $X_1$ and $X_2$. A representative fitted spectrum is shown in Fig. 6d, where the individual contributions are also displayed. To produce the left panel on Fig. 6g, the different contributions were multiplied by a spectrum representing the bandpass grating filter used experimentally, and the $X_1$ contribution was then normalized to the sum of the integrated intensities of all components before filtering. The wavelength detuning $\Delta$ between $X_1$ and the cavity was determined from these fits. All uncertainties quoted for $\Delta$ and the $X_1$, $X_2$ and cavity contributions correspond to 95% fit confidence intervals (two standard deviations).

**Photoluminescence decay measurements.** For excited state lifetime measurements, we employed a 10 GHz lithium niobate EOM to produce a 80 MHz, ≈200 ps pulse train from the CW ECDL laser. A fiber-based polarization controller was used to control the polarization of the ECDL light going into the EOM, and a DC bias was applied to the EOM to maximize signal extinction. An electrical pulse source was used to produce an 80 MHz train of ≈200 ps pulses of <1 V peak amplitude, which was then amplified and used to drive the EOM via its radio frequency port. A trigger signal from the pulse generator served as the reference channel in our TCSPC system. Supplementary Fig. 6a shows a typical temporal profile for the pulses produced by the EOM, detected with an SNSPD. Pulse FWHM of ≈200 ps and >20 dB extinction are observed. The pulsed electrical signal produced small satellite peaks that were imprinted in the optical signal, as indicated in Supplementary Fig. 6a. These satellite peaks typically appeared a few ns after each proper pulse, and were ≈20 dB below the latter in intensity. Impulse response functions (IRFs) such as the one in Supplementary Fig. 6a were used in decay lifetime fits as explained below, so that the effect of satellite peaks, though minimal, was accounted for. Finally, to determine the time resolution of our detection system, we launched attenuated few-ps pulses from a Ti:Sapphire mode-locked laser at 975 nm into the SNSPDs, to obtain the temporal trace in Supplementary Fig. 6b. The peak can be well fitted with a Gaussian with standard deviation $\sigma =$ 129 ps ± 0.04 ps (uncertainty is a 95% least-squares fit confidence interval, corresponding to two standard deviations).

**Photoluminescence decay fits.** QD emission decay fits were performed using maximum likelihood estimation. We consider a lifetime trace $Y^k = \{Y_i\}_{i=1}^k$ where a known number of photon counts $N$ is distributed over $k$ time bins, such that the bin counts $y_i$ follow a multinomial distribution[43]. The maximum likelihood estimator is

$$g_{\text{MLE}}(y^k) = \arg\min_{\theta \in \Theta} \left\{ -\sum_{i=1}^{k} y_i \ln p_i(\theta) \right\}, \qquad (8)$$

where $\theta$ is a vector in the multidimensional parameter space $\Theta$. Estimates for the various fit parameters are obtained by finding $\theta$ that minimizes the expression in the curly brackets, where $y_i$ is the $i$-th bin count, and $p_i(\theta)$ is a probability density function that models the decay, evaluated at the $i$-th bin. We define $p_i(\tau) = e^{-ir/k}\frac{e^{\frac{r}{k}}-1}{1-e^{-r}}$, with $r = \frac{i \cdot \Delta t}{\tau}$. For a monoexponential decay when a portion $b$ of the signal is due to background emission,

$$p_i(\theta) = p_i(\tau, b) = \frac{b}{k} + (1-b)p_i(\tau) \qquad (9)$$

For biexponential decay with a background $b$, let $\tau \stackrel{\Delta}{=} (\tau_1, \tau_2)^T$. Then $p_i(\tau, a, b)$ (where $a$ is the contribution of the first exponential decay) may be expressed as

$$p_i(\theta) = p_i(\tau, b, a) = \frac{b}{k} + (1-b)[ap_i(\tau_1) + (1-a)p_i(\tau_2)] \qquad (10)$$

Variances for the estimated parameters in $\theta$ can be obtained from the diagonal elements of the inverse of the Fisher Information Matrix (Supplementary Note 8). In the fitting procedure, the trial decay function $p_i(\theta)$ is numerically convolved with the experimentally measured, background-subtracted IRF and used in Eq. (8). Because the optical pulses used to obtain the IRF follow a considerably different path length towards the detector than the QD signal, the IRF and QD decay traces are delayed with respect to each other. We manually align the two traces to minimize fit residuals. Uncertainties given in the text correspond to standard deviations for the various parameters, obtained from the diagonal elements of the inverse of the Fisher information matrix computed with the expectation values from the fit (corresponding to the Cramér–Rao lower bound).

**Estimate of $\beta$.** Below we estimate the coupling $\beta$ of the QD exciton at $\lambda \approx$ 1330.18 nm of Fig. 5a into the guided TE mode of the GaAs waveguide where it was hosted. Ideally such a measurement would involve saturating the QD under pulsed excitation, where the maximum possible photon flux from the QD is given by the laser repetition rate. Because a pulsed source with sufficient power to saturate the QD was unavailable, our estimate relied on the continuous-wave emission spectrum of Fig. 4a. A three-level system model for the QD was then used to account

for blinking. First, we measured the spectrum of a laser signal of known power at 1070 nm with our spectrometer, using the same fiber-coupled input as that for Fig. 5a. The laser was attenuated with a calibrated variable optical attenuator, and launched into a fiber-based 10:90 power splitter (with a calibrated power-splitting ratio), the 90% port of which was sent to a photodiode for power monitoring. Integration of the background-subtracted laser spectrum counts divided by the laser power gave a factor of 0.0023 counts per photon at the spectrometer fiber-coupled input (this includes losses at the fiber connector, spectrometer slit, grating and output slit before the InGaAs detector array). This allowed us to obtain, from the fitted QD spectrum of Fig. 5a, a photon flux $P = 3.0 \times 10^6\,\text{s}^{-1} \pm 0.5 \times 10^6\,\text{s}^{-1}$ (errors come from the 95% fit confidence intervals) at this fiber input for the 1130.18 nm exciton line (accounting for the wavelength difference). We next expanded the photon flux as $P = X\beta\eta\eta_{\text{ext.}}\eta_{\text{TF}}$, where $X$ is the exciton population probability, $\eta$ the mode transformer efficiency, $\eta_{\text{ext.}}$ the lensed fiber-to-chip coupling efficiency, and $\eta_{\text{TF}} = 0.91 \pm 0.03$ is the lensed fiber transmission (uncertainty from measurement error, corresponding to one standard deviation). Solving the three-level system rate equations (with one bright and one dark transition) that fit the $g^{(2)}(\tau)$ data in Fig. 4c—assuming the lifetime in Fig. 4d for the bright transition —we obtain $X = 0.15 \pm 0.04$, where the uncertainty is the 95% fit confidence interval. We note that connecting the dark state to either the ground or bright excited state in our model leads to $X \approx 0.15$. Assuming $\eta = 98\%$ (the maximum from simulation) and $\eta_{\text{ext.}} = 0.22$, a reasonable value from Supplementary Fig. 8b at 1130 nm, we obtain, propagating uncertainties, $\beta = 0.20 \pm 0.07$.

**Data availability**. All data supporting this study are openly available from the University of Southampton repository at http://doi.org/10.5258/SOTON/D0174.

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

## Acknowledgements

We thank Daron Westly and Rob Ilic from the NIST CNST for invaluable aid with fabrication, and Raphael Daveau from the Niels Bohr Institute, University of Copenhagen, Denmark, for aid with the optical setup. We also thank A. Stintz and S. Krishna from the University of New Mexico for providing the quantum dot epitaxy used in this work. J.L. acknowledges the Ministry of Science and Technology of China (grant no. 2016YFA0301300), the National Natural Science Foundation of China(grant no. 11304102), and support under the Cooperative Research Agreement between the University of Maryland and NIST-CNST, Award 70NANB10H193. L.S. acknowledges financial support from EPSRC, grant EP/P001343/1. J.V.D.M.C. acknowledges funding from the Brazilian Ministry of Education through the Brazilian Scientific Mobility Program CAPES-grant 88888.037310/2013-00.

## Author contributions

M.D. developed computational electromagnetics models and designed the devices, developed the fabrication process, designed and performed the experiments, analyzed the data, and wrote the manuscript. J.L. produced wafer-bonded samples, performed experiments and wrote the manuscript. L.S. performed experiments and wrote the manuscript. C.-Z.Z. and L.L. designed devices and produced wafer-bonded samples. J.V.D.M.C. wrote the lifetime fitting scripts. V.V., R.M., and S.W.N. provided the nanowire superconducting single-photon detectors. K.S. supervised the project and wrote the manuscript.

## Additional information

**Competing interests:** The authors declare no competing financial interests.

