## [Peer Review File · Nature Communications]

Reviewers' Comments:

Reviewer #1:

Remarks to the Author:

I am happy to recommendation publication for this new submission. The authors have revised the earlier submission. It meets the high standard of nature communications.

Reviewer #2:

Remarks to the Author:

I had four comments on the paper, and the authors have address them well in a point by point manner. Therefore, I think that the paper can be published in Nature Communications, especially due to the potential of scalability for silicon-based quantum photonic circuits.

Below are the comments from the two reviewers regarding our manuscript NCOMMS-17-12692-T. We thank both reviewers for the careful reading of our manuscript, and the very positive comments about our work.

REVIEWERS' COMMENTS:

Reviewer #1 (Remarks to the Author):

I am happy to recommend publication for this new submission. The authors have revised the earlier submission. It meets the high standard of nature communications.

Reviewer #2 (Remarks to the Author):

I had four comments on the paper, and the authors have address them well in a point by point manner. Therefore, I think that the paper can be published in Nature Communications, especially due to the potential of scalability for silicon-based quantum photonic circuits.